# Lamin A/C-Dependent Translocation of Megakaryoblastic Leukemia-1 and β-Catenin in Cyclic Strain-Induced Osteogenesis

**DOI:** 10.3390/cells10123518

**Published:** 2021-12-14

**Authors:** Asmat Ullah Khan, Rongmei Qu, Yuchao Yang, Tingyu Fan, Yan Peng, Bing Sun, Xianshuai Qiu, Shutong Wu, Zetong Wang, Zhitao Zhou, Muhammad Akram Khan, Jingxing Dai, Jun Ouyang

**Affiliations:** 1Guangdong Provincial Key Laboratory of Medical Biomechanics & Guangdong Engineering Research Center for Translation of Medical 3D Printing Application & National Key Discipline of Human Anatomy, School of Basic Medical Sciences, Southern Medical University, Guangzhou 510515, China; khan9905@smu.edu.cn (A.U.K.); shirth@smu.edu.cn (R.Q.); yangyuchao@smu.edu.cn (Y.Y.); fantingyu@i.smu.edu.cn (T.F.); pengyan0470@i.smu.edu.cn (Y.P.); xing19951222@i.smu.edu.cn (B.S.); sylviawoo@i.smu.edu.cn (S.W.); 2Department of Spine Surgery, Zhujiang Hospital, Southern Medical University, Guangzhou 510280, China; qiuxs@i.smu.edu.cn; 3The First Clinical Medicine College, Southern Medical University, Guangzhou 510515, China; 3170011034@i.smu.edu.cn; 4Central Laboratory, Southern Medical University, Guangzhou 510515, China; zzt2010@smu.edu.cn; 5Department of Veterinary Pathology, Faculty of Veterinary and Animal Sciences, PMAS-Arid Agriculture University, Rawalpindi 46300, Pakistan; dr.m.akram@uaar.edu.pk

**Keywords:** β-catenin, megakaryoblastic leukemia-1 (MKL1), lamin A/C (LMNA), mechanical strain, osteogenesis

## Abstract

Lamins are intermediate filaments that play a crucial role in sensing mechanical strain in the nucleus of cells. β-catenin and megakaryoblastic leukemia-1 (MKL1) are critical signaling molecules that need to be translocated to the nucleus for their transcription in response to mechanical strain that induces osteogenesis. However, the exact molecular mechanism behind the translocation of these molecules has not been fully investigated. This study used 10% cyclic strain to induce osteogenesis in the murine osteoblast precursor cell line (MC3T3). The translocation of β-catenin and MKL1 was studied by performing knockdown and overexpression of lamin A/C (LMNA). Cyclic strain increased the expression of osteogenic markers such as alkaline phosphatase (ALP), runt-related transcription factor 2 (RUNX2), and enhanced ALP staining after seven days of incubation. Resultantly, MKL1 and β-catenin were translocated in the nucleus from the cytoplasm during the stress-induced osteogenic process. Knockdown of LMNA decreased the accumulation of MKL1 and β-catenin in the nucleus, whereas overexpression of LMNA increased the translocation of these molecules. In conclusion, our study indicates that both MKL1 and β-catenin molecules are dependent on the expression of LMNA during strain-induced osteogenesis.

## 1. Introduction

The nuclear lamina is a type VI intermediate filaments meshwork found inside the nuclear envelope. In addition to its role as a structural barrier between cytoplasm and nucleus, it plays an essential part in several cellular processes. Two functions are usually assigned to the nuclear lamins: (1) it provides shape and structural support to the nucleus; (2) it regulates cell signaling and chromatin remodeling. Recently, a few studies have suggested the role of nuclear lamin in sensing mechanical forces and acting as a mediator to generate a mechanical response to adapt to the changing microenvironment [1,2].

Two isoforms of nuclear lamins, generated by splicing lamin A/C (LMNA) genes, i.e., lamin type A and C, are highly expressed in differentiated cells [3]. Several laminopathies have been reported based on their LMNA mutation. For example, LMNA mutation leads to Hutchinson–Gilford progeria syndrome (HGPS), characterized by lipodystrophy, aging, joint contracture, and osteoporosis [4]. In vivo studies have suggested that cells with HGPS have a nuclear defect, and their mechanosensing ability is also altered [5,6]. In another disease, Mandibular Dysplasia Type A, a mutation in the LMNA leads to osteolysis and osteoporosis [7]. In atypical progeroid syndrome, age-related pathologies such as osteoporosis and thinning of the cortical structure occurred due to defects in the LMNA gene [8,9]. However, the exact mechanism of laminopathies has not been studied. One hypothesis is that lamin mutation causes a disruption in cell signaling and gene expression, which is also supported by a study in which fibroblast cells showed an interrupted megakaryoblastic leukemia-1/Serum response factor (MKL1/SRF) signaling in LMNA-depleted cells [10].

The role of LMNA in osteogenesis has attracted attention, since it plays a significant role in laminopathies [11,12]. A study on mice observed age-related bone loss in four-week-old mice because they were deficient in the LMNA gene [13,14]. Moreover, the knockdown of the LMNA in mesenchymal stem cells (MSCs) resulted in reduced osteogenesis and runt-related transcription factor 2 (RUNX2) expression. Similarly, LMNA overexpression increased the osteogenesis process by regulating the Notch pathway [15]. Substrate stiffness also affected the LMNA expression and its effect on differentiation. For example, the soft substrate with knockdown LMNA gene favored adipogenesis, whereas a stiff substrate with a regulated LMNA gene facilitated osteogenesis [16]. These results suggest that LMNA could be a contributing factor in stress-induced osteogenesis.

Mechanical strain plays an essential role in regulating bone regeneration, since increased mechanical strain causes osteoblast proliferation and differentiation [17,18]. One study indicated that high-frequency vibration augmented the osteogenic differentiation in bone marrow stem cells (BMSCs) [19]. Moreover, low amplitude high-frequency was reported to increase the bone formation process in the initial stages [20]. Similarly, the cyclic strain also increased extracellular matrix mineralization through extracellular signal-regulated kinase (ERK1/2) human mesenchymal stem cells [21]. Several other studies have also indicated the increased osteogenic differentiation in human-derived adipose stem cells (hASCs) [22,23], BMSC [24,25,26], human intraoral mesenchymal stem and progenitor cells [27], human embryonic stem cells (hESCs) [28], human fibroblast [29], and MC3T3-E1 cells [30,31] under the influence of cyclic tensile stress. In comparison, no response to reduced bone formation was observed without mechanical loading [32,33,34]. During mechanical induced osteogenesis, mechanosensors sense the strain and transfer the signal into the nucleus. It has also been reported that A-type lamins are the leading key player responsible for the mechanotransduction-induced cell pathways. Lamin acts as a molecular sensor and plays a vital role in transducing mechanical signals into biological signals involved in cell survival, growth, differentiation, and disease progression [16,35,36]. However, the molecular mechanism of stress-induced osteogenesis remains to be identified. 

Previous reports have established that Wnt/β-catenin and MKL1/SRF pathways are regulatory factors for osteogenesis [37,38,39]. β-catenin, being an essential part of adherent junction, regulates cell adhesion and activates Wnt cell signaling. The nuclear translocation of β-catenin causes the activation of specific genes through transcription factors and mediates cellular development [40,41]. Consequently, the activation of Wnt/β-catenin signaling results in the differentiation of mouse mesenchymal stem cells [42]. Similarly, the translocation of MKL1 is mainly regulated by the actin cytoskeleton, which is coupled with lamin A/C by a linker of nucleoskeleton and cytoskeleton (LINC) complexes [43]. The state of actin depolymerization results in higher G-actin monomers, which binds with MKL1 and help its translocation in the nucleus for the target genes [44]. MKL1/SRF activation also results in stress fibers formation, essential for mechanotransduction [43]. Furthermore, it is also reported that both MKL1 and β-catenin are mechanosensitive signaling molecules that respond to the mechanical strain [38,45]. During chemical-induced osteogenesis, LMNA causes translocation of β-catenin into the nucleus [46]. Moreover, LMNA regulates the translocation of MKL1 in cardiomyocytes [10]. Thus, we hypothesize that, during the mechanical activation of osteogenesis, the translocation of MKL1 and β-catenin are dependent on LMNA.

## 2. Results

### 2.1. Cyclic Strain and Osteogenic Differentiation

MC3T3-E1 cells were cultured on FlexCells culture plates, and 10% cyclic strain was applied on the 1st, 4th, and 7th days (Figure 1A). Western blot and RT-qPCR analysis indicated that osteogenic markers, i.e., RUNX2 and ALP (Alkaline phosphatase), gradually increased and were statistically significant on the 7th day compared to the 0 and 1st days (Figure 1B,C). ALP staining also increased on the 7th day, suggesting the activation of osteogenesis due to mechanical strain (Figure 1D). These results indicated that cyclic strain promoted osteogenesis in MC3T3-E1 cells.

Furthermore, lamin A/C protein expression peaked at day seven compared to control. The immunofluorescence expression of LMNA also became prominent on the 4th and 7th days after applying cyclic strain (Figure 1E). Similarly, the expression of the signaling molecules MKL1 and β-catenin upregulated significantly during this period and was highest on the 7th day (Figure 1B,C). Overall, it indicated that osteogenic induction was encouraged under the influence of cyclic strain, and LMNA, MKL1, and β-catenin were upregulated during strain-induced osteogenesis.

### 2.2. Cyclic Strain and Translocation of Signaling Molecules

After investigating the protein expression during the cyclic strain, we observed the translocation of β-catenin and MKL1 signaling molecules in the cytoplasm and nucleus. A cytoplasmic and nuclear extraction kit was used, wherein samples were collected immediately after applying strain. The result showed that MKL1 gradually decreased in the cytoplasm and reached its lowest level on the 7th day. Conversely, translocation in the nucleus increased slowly and reached to peak on the 7th day. Similarly, β-catenin also decreased gradually over time in the cytoplasm, and its accumulation significantly increased in the nucleus by following the same trend (Figure 2A,B). Immunofluorescence of samples without stretch (before application of strain) and after stretch (after the cyclic strain) revealed that MKL1 expression was reduced on days 4 and 7 without stretch. However, increased nuclear accumulation was observed on the 4th and 7th days after the stretch (after cyclic strain). Similarly, β-catenin also increased nuclear translocation on the 4th and 7th days (Figure 3A,B). Both MKL1 and β-catenin retained their nuclear expression even after 8 h of stimulation (Appendix A). Overall, results indicated that both MKL1 and β-catenin were translocated in the nucleus after applying strain. 

### 2.3. Knockdown and Overexpression of LMNA

As we observed increased LMNA expression and increased nuclear accumulation of MKL1 and β-catenin, we speculated that LMNA might influence the translocation of these molecules during the process. We then knocked down and overexpressed LMNA to investigate its effect on the translocation of signaling molecules. Sh plasmid and lentivirus were used for the knockdown and the overexpression of LMNA, respectively. Fluorescence microscopy was used to observe green fluorescent protein (GFP), which showed that transfection efficiency was above 80% (Figure 4A). Both knockdown and overexpression LMNA were then confirmed using Western blot, RT-qPCR, and immunofluorescence (Figure 4B). Interestingly, phalloidin staining revealed more distorted actin filaments in the LMNA knockdown cells over the nucleus, whereas abundant stress fibers were visible in control and overexpressed LMNA cells (Figure 4B). Moreover, ALP and alizarin staining were performed on knockdown and overexpressed LMNA cells (Figure 1C,D), which showed significantly higher staining in the overexpressed LMNA (OE) cells as compared to the LMNA knockdown (KD) and the control (CTL) cells (Figure 1C,D).

The cyclic strain was applied for seven days, and the cytoplasmic and nuclear protein were collected on the 7th day. Knocking down LMNA reduced total MKL1, RUNX2 expression significantly; however, its effect on β-catenin was insignificant (Figure 5A,B). It indicated that LMNA knockdown decreased strain-induced osteogenesis. Next, we observed the difference in the translocation of these molecules using nuclear and cytoplasmic protein and immunofluorescence. We found that translocation of MKL1 and β-catenin was significantly affected by knocking down LMNA. It indicated that both MKL1 and β-catenin in knockdown LMNA cells show reduced accumulation in the nucleus compared to control (Figure 5C).

We then investigated whether increased LMNA expression causes increased nuclear accumulation. LMNA overexpression caused a significant rise in the expression of β-catenin, RUNX2, and MKL1. Besides, both MKL1 and β-catenin were prominently expressed in the nucleus. Moreover, immunofluorescence confirmed the Western blot results, as MKL1 and β-catenin were highly expressed in the nucleus in overexpressed LMNA cells (Figure 6A,B). These results indicated that knocking down LMNA decreased protein expression and decreased nuclear translocation of β-catenin and MKL1. Conversely, LMNA overexpression upregulated the MKL1, RUNX2, and β-catenin, and increased the nuclear translocation of MKL1 and β-catenin (Figure 6C and Figure 7A,B).

## 3. Discussion

Mechanotransduction initiates the movement of mechanosensitive molecules and transcription factors in the nucleus during and after the mechanical force [47,48]. The activation of mechanotransduction stimulates bone formation, thus keeping the balance between bone absorption and bone formation [49]. Among a few mechanosensors, LMNA is an intermediate filament lying inside the nuclear envelope that transfers the mechanical signals into the nucleus and adapts to the changing mechanical environment [50,51]. In this study, LMNA dependent translocation of two important signaling molecules, i.e., MKL1 and β-catenin in the nucleus, was studied under the mechanical strain. 

The cyclic mechanical strain has been reported to induce osteogenesis in osteoblast cell lines and mesenchymal stem cells [52,53]. In our study, cyclic strain increased osteogenic markers such as RUNX2 and ALP on days 1 and 4, which became highest on day 7. These results are consistent with other studies that reported that a 10% cyclic strain increased RUNX2 and ALP expression after applying strain [17,54]. The upregulation of LMNA could be explained by the evidence that A-type lamins are the main filaments that respond to mechanical strain. Soft substrate results in wrinkled nuclei, and stiff substrate causes nuclei to be stiffer and straight [35]. Moreover, higher lamin quantity results in the increased stiffness of the matrix itself. The higher degree of lamin stiffness might be attributed to the increased translocation of transcription factors such as MKL1 and YAP [16,55,56]. Furthermore, an increased β-catenin and MKL1 expression under mechanical strain were also reported in a few studies [49,57,58].

Translocation of β-catenin and MKL1 in the nucleus reached a peak on the 4th and 7th days. Our study demonstrated that β-catenin increased in the cytoplasm on the first day and then decreased gradually, which is supported by the fact that β-catenin is a mechanosensitive signaling molecule bound to cadherin mechanosensor. After the application of strain, β-catenin is released free from cadherin in the cytoplasm [59]. The translocation of β-catenin to the nucleus is needed, since it activates the transcription of downstream factor RUNX2 [42]. MKL1 is another mechanosensitive protein bound to G-actin in the cytoplasm, which prevents its translocation into the nucleus [55,60]. Mechanical strain induces actin polymerization and stress fibers formation, which enables the release of MKL1 and translocates it to the nucleus [61,62]. MKL1 is reported to inhibit the PPARγ in the nucleus [62,63]. The possible cross-talk between MKL1 and β-catenin could also explain their same translocation trend under the cyclic strain. For example, Smad-3, a potent MKL1 transcription inhibitor, competes with β-catenin, and the association of Smad-3 with β-catenin results in freeing MKL1, which can bind to SRF [64]. Moreover, MKL1 is degraded when Smad-3 recruits glycogen synthase kinase-3β; however, increased β-catenin suppresses this recruitment process, thus leading to the stability of MKL1 [65]. Detailed investigation of their interaction needs to be explored in potential future studies.

In our study, alizarin staining was more evident in LMNA overexpressed cells, indicating more osteogenesis in LMNA overexpressed cells than in LMNA knockdown and control. LMNA knockdown decreased the MKL1, RUNX2 expression, and its overexpression increased MKL1, β-catenin, and RUNX2 expression. This could be supported by the fact that MKL1 decreased its response in MSCs when cultured on the soft substrate, and its expression increased when cultured on the stiff substrate [38]. In our study, nuclear translocation was also minimum in LMNA knockdown cells under the mechanical strain. This evidence also supports the fact that, in LMNA deficient fibroblast cells, most MKL1 were translocated in the cytoplasm, leaving behind very little in the nucleus [10,66]. Matrix stiffness also increased lamin expression and induced MKL1 translocation in the nucleus [16]. Interestingly, nuclear β-catenin was expressed less compared to control in knockdown LMNA cells in our study. This might be because the nuclear import of protein is reduced in LMNA deficient cells [67]. Bermeo et al. [46] reported cytoplasmic accumulation of β-catenin in LMNA deficient cells during chemical-induced osteogenesis. On the other hand, our study reported that overexpressed LMNA cells increased RUNX2, MKL1, and β-catenin expression. Moreover, LMNA overexpression resulted in a higher amount of actin fibers and knockdown resulted in actin degradation, especially around the nucleus. Actin polymerization might be the reason for MKL1 translocation in the cytoplasm because LMNA and Emerin regulate actin polymerization, which can cause increased MKL1 translocation in the nucleus.

In summary, we present the first evidence that translocation of MKL1 and β-catenin is dependent on LMNA under the cyclic strain’s induced osteogenesis. However, we think there are some limitations in our study, i.e., firstly, the study duration is seven days, which could only tell us about the initial stage of mechanical induced osteogenesis. Secondly, we observed the MKL1 and β-catenin translocation immediately after applying strain; however, translocation could also be studied on different time scales after two hours or four hours, which helps us give more insight into the translocation of transcription factors. Overall, these data suggest that MKL1 and β-catenin are dependent on LMNA for their translocation in the nucleus under the mechanical strain. Future studies will focus on cross-talk between MKL1 and β-catenin under different time durations and frequencies of mechanical strain.

## 4. Materials and Methods

### 4.1. Cell Culture

MC3T3-E1 cells were purchased from Chinese Academy Sciences cell back bank (ATCC CRL-2594, Shanghai, China). MC3T3-E1 cells were cultured in MEM alpha (Gibco, Grand Island, NY, USA) at 37 °C in a 5% CO_2_ atmosphere. The medium was supplemented with 10% FBS (Gibco) and 1% penicillin and streptomycin (Gibco). The medium was changed regularly every second day to maintain the working conditions.

### 4.2. Cyclic Strain Loading

Cells were cultured at the confluence of 1 × 10^5^ cells on specialized collagen-coated silicone membrane plates (Bioflex, Flexcell International, Burlington, NC, USA). The cyclic strain was applied after 48 h of culturing MC3T3-E1 cells on the plate. Flexcell FX-5000 system (Flexcell International) (Figure 1A) was used to apply the mechanical strain on the collagen-coated plates with MC3T3-E1 cells cultured on them. The intensity of cyclic strain was 10% tensile stress with a frequency of 0.5 HZ, which was delivered for 2 h every day. Protein and RNA were extracted from the cells immediately after the application of strain. Cells cultured with a similar condition but without stretching served as control.

### 4.3. ALP Staining

Phosphate buffer saline (PBS) was used to wash the cells, and then 4% paraformaldehyde was used to fix the cells. PBS was used to wash again, and ALP staining was used (Beyotime, Shanghai, China). Cells were incubated for 30 min at 37 °C and then washed twice with PBS. An optical microscope was used to observe the cells after washing with PBS.

### 4.4. Alizarin Staining

Culture medium was removed from the plates, and PBS was used to wash the cells twice. Fixation was carried out using 4% paraformaldehyde followed by washing of the cells with PBS. Cells were incubated with alizarin staining (Cyagen, Suzhou, China) for 10 min and then washed again with PBS two times. Stained cells were observed under the microscope (1MT-2-21, Olympus, Tokyo, Japan).

### 4.5. Immunofluorescence

Followed by washing with PBS three times, cells were fixed with 4% paraformaldehyde (Solarbio, Beijing, China) for 10 min at room temperature. PBS was used to wash these cells after fixation, and 0.1% Triton X-100 was used for 5 min. Then, blocking was performed by 5% bovine albumin serum for 30 min at room temperature. Cells were incubated overnight at 4 °C with primary antibodies (mouse anti-Lamin A/C (1:500, ab8984, Abcam, London, UK), Rabbit anti-MKL1 (1:500, ab49311, Abcam), rabbit anti-β-catenin (1:500, D10A8, Cell Signaling Technology, Danvers, MA, USA), and Phalloidin Red-594 (1:500, Beyotime, C2203S, Shanghai, China)). Cells were again washed three times, and the second antibodies (AlexaFluor488 goat anti-mouse (1:500), AlexaFluor488 goat anti-rabbit (1:500), AlexaFluor568 goat anti-mouse (1:500), AlexaFluor568 goat anti-rabbit (1:500), (Invitrogen, USA)) were added. Cells were incubated for 1 h in the dark. These cells were imaged under confocal microscopy, and the data were analyzed using the SPSS^®^ Statistics 20 software.

### 4.6. Cytoplasmic and Nuclear Protein Extraction

After applying mechanical strain, cells were harvested immediately on days 1, 4, and 7. Cytoplasmic Protein Extraction Kit (Keygen Biotech, Nanjing, China) was used, and protein extraction was done according to the manufacturer’s instructions. According to the provided protocol, the cytoplasmic protein was extracted first, followed by the nuclear protein extraction.

### 4.7. Western Blotting

After harvesting and washing the cells, the protein was extracted by preparing immunoprecipitation lysis buffer using the instruction given in the whole protein extraction kit (Key GEN BioTECH, Nanjing, China). A total of 10% SDS polyacrylamide gel electrophoresis was used to separate the protein and then transfer it onto polyvinylidene difluoride (PVDF) membranes. After the transfer, blocking was carried out by 5% skim milk for one hour. All membranes were incubated at 4 °C overnight in the primary antibody mouse anti-Lamin A/C (1:1000, Abcam, ab8984), Rabbit anti-MKL1 (1:1000 Abcam ab49311), rabbit anti-β-catenin (1:1000, Cell Signaling Technology, D10A8) mouse anti-ALP (1:1000, Abcam, ab126820), rabbit anti-RUNX2 (1:1000, Cell Signaling Technology, 12556S), and rabbit anti-GAPDH (glyceraldehyde-3-phosphate dehydrogenase) (1:5000, Bioworld, ap0063). After that, membranes were washed and incubated with a second antibody at room temperature for two hours. Enhanced chemiluminescence (ECL) chromogenic substrate was used to observe the band.

### 4.8. Quantitative Real-Time Polymerase Chain Reaction (RT-qPCR)

Trizol (Invitrogen, Carlsbad, CA, USA) was used to extract the total RNA from the cells following the instructions given by the company. RevertAid First Strand cDNA Synthesis Kit (Thermo Fisher, Waltham, MA, USA) was used to reverse-transcribed RNA into cDNA. ABI StepOne Plus System (American Applied Biology Systems Inc., Waltham, MA, USA) was used to perform a quantitative real-time polymerase chain reaction (qPCR). Following primer sequences were used to express LMNA, β-catenin, RUNX2, ALP, MKL1 Table 1.

### 4.9. Lentivirus Transduction for Overexpression

Lentivirus for negative control (CON335) and LV-LMNA (66639) constructed by Genechem (Shanghai, China) were used to trigger overexpression of LMNA. First, the optimal condition was selected, including optimal concentration, volume, treatment duration, and the number of cells using multiple treatments. After that, virus dilution in the fresh medium according to transfected cell density ratio and virus efficiency enhancer were added. Firstly, approximately 7000 cells/cm2 were cultures in 6-well culture plates. When the confluence reached 20 to 30 percent, we transfected the virus into the cells using the cell multiplicity of infection (MOI) and virus titer. The medium was replaced with a growth medium after 12 h, and expression efficiency was observed under an Olympus microscope (Tokyo, Japan).

### 4.10. Knockdown LMNA

Cells were grown in 6-well plates, and the growth medium was supplemented until it reached the confluence of 70%. LMNA Sh-RNA plasmid (Santacruz Biotechnology, sc-293885) were diluted in transfection medium (Santacruz Biotechnology, sc-108062) and incubated for 30 min at room temperature. Cells were washed with transfection medium, and plasmid was inoculated for 7 h. The medium was aspirated after 7 h, and the growth medium was transferred into it. Western blot and RT-qPCR experiments were performed to confirm the knockdown.

## Figures and Tables

**Figure 1 cells-10-03518-f001:**
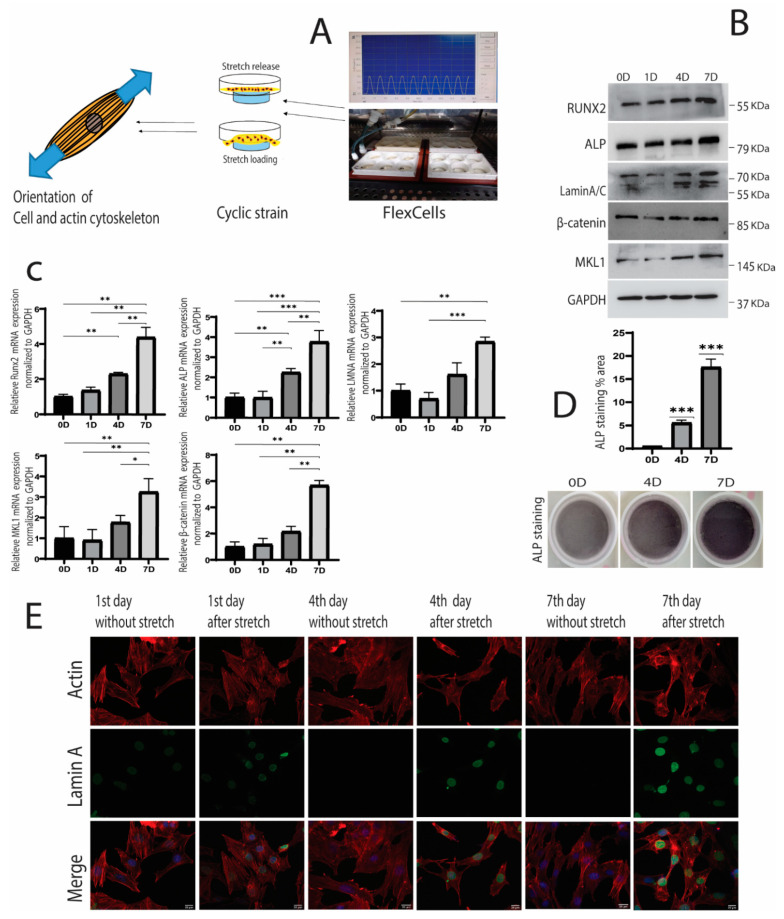
(**A**) The FX-5000 system (Flexcell International), the direction of cyclic strain, and the orientation of actin cell cytoskeleton. (**B**) The protein expression of β-catenin, megakaryoblastic leukemia-1 (MKL1), alkaline phosphatase (ALP), runt-related transcription factor 2 (RUNX2), and lamin A/C (LMNA) in MC3T3-E1 cells under cyclic strain. β-catenin, MKL1, ALP, RUNX2, LMNA protein expression increased gradually and peaked at 7th day (Western blot analysis); (**C**) The relative mRNA upregulated expression of β-catenin, MKL1, ALP, RUNX2, and LMNA (Lamin A/C) in MC3T3-E1 cells under cyclic strain; (**D**) Quantification of ALP staining was performed using NIH image J, * *p* < 0.05, ** *p* < 0.01, *** *p* < 0.001, *n* = 3; ALP staining on day 0, 4, and 7 showed increased staining at 7th day as compared to day 0; (**E**) The immunofluorescence expression of lamin-A in MC3T3-E1 cells under cyclic strain and without strain. Scale bar = 20 µm.

**Figure 2 cells-10-03518-f002:**
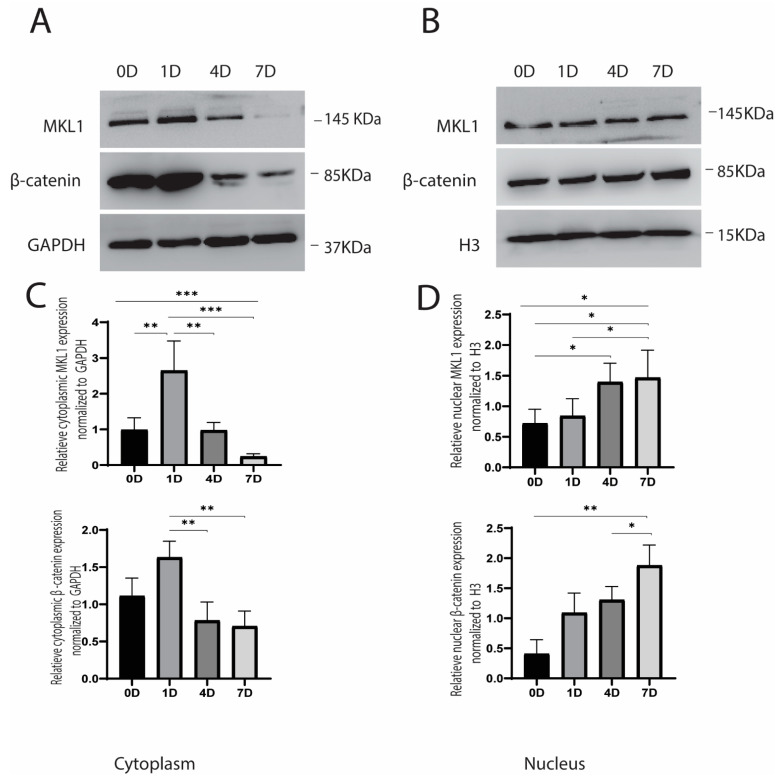
(**A**) The protein expression of MKL1 and β-catenin decreased significantly in the cytoplasm on days 4 and 7 (Western blot analysis); (**B**) The protein expression of MKL1 and β-catenin increased significantly in the nucleus on day 7 (Western blot analysis); (**C**) The graph shows that MKL1 increased on the first day in the cytoplasm and gradually decreased; (**D**) The graph shows that MKL1 and β-catenin increased gradually in the nucleus and peaked on the 7th day. Densitometric quantification of the Western blotting bands normalized to Glyceraldehyde 3-phosphate dehydrogenase (GAPDH) for cytoplasmic protein and Histone (H3) for nuclear protein * *p* < 0.05, ** *p* < 0.01, *** *p* < 0.001, *n* = 3.

**Figure 3 cells-10-03518-f003:**
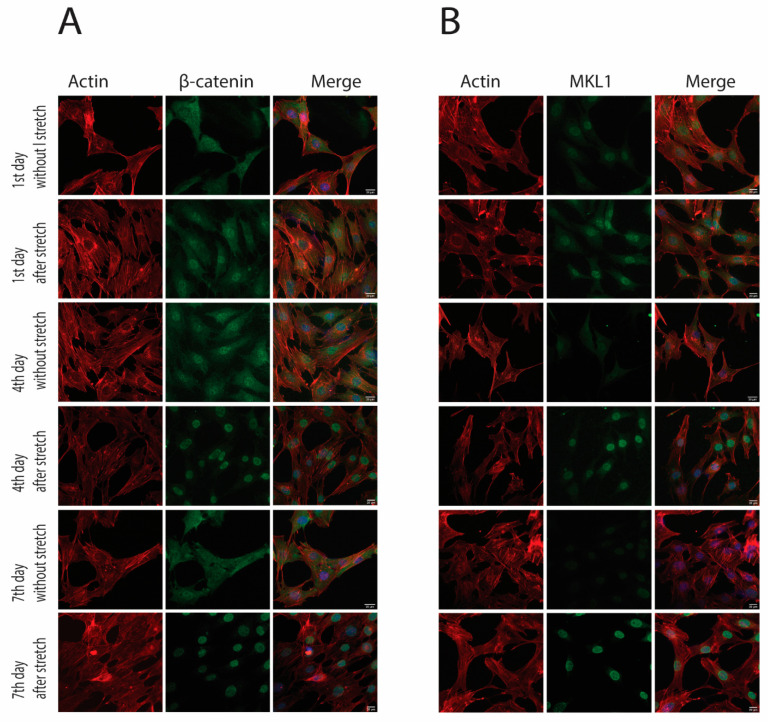
The immunofluorescence expression of β-catenin and MKL1 in MC3T3-E1 cells after a stretch and without stretch wherein stress fibers aligned in an angle to the direction of strain. (**A**) β-catenin expression was cytoplasmic and nuclear in control and day 1, but it indicates nuclear accumulation on the 4th and 7th days (**B**). The expression of MKL1 was more cytoplasmic on the control and the 1st day, but it showed higher nuclear translocation on the 4th and 7th days. Scale bar = 20 µm.

**Figure 4 cells-10-03518-f004:**
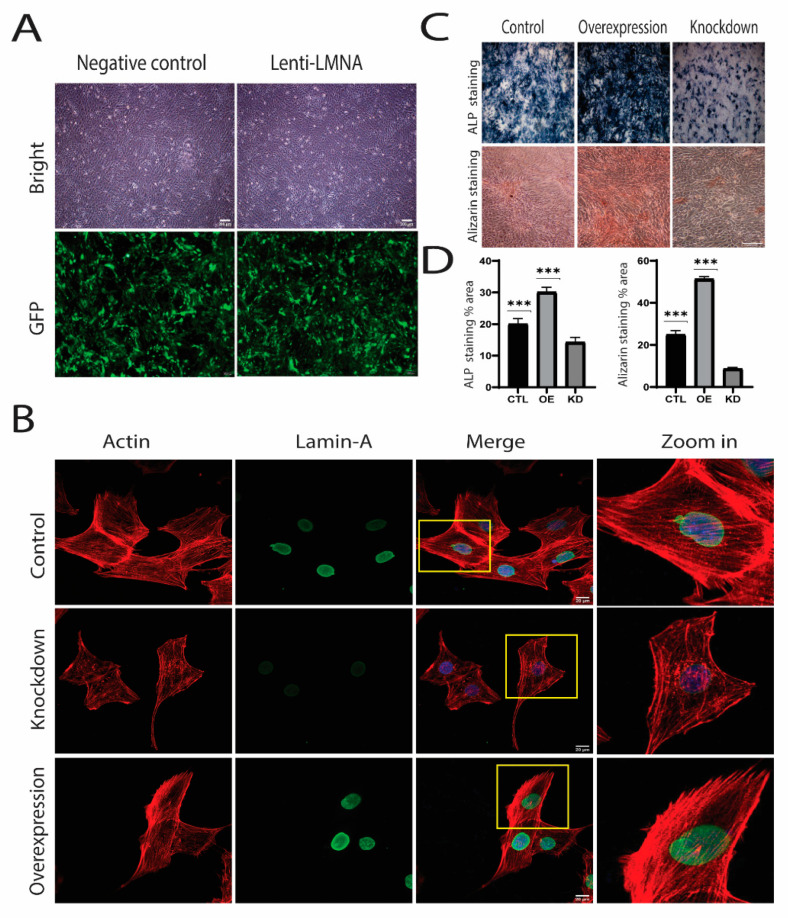
(**A**) MC3T3-E1 fluorescence intensity five days after virus infection. Scale bar = 200 µm; (**B**) The expression of knockdown (KD) and overexpression (OE). LMNA was detected by immunofluorescence. Scale Bar = 20 µm. Zoom in pictures showed that knockdown LMNA lacked actin fibers while overexpression showed intense phalloidin staining; (**C**) Alizarin staining showed that overexpression indicated more mineralization than control (CTL), and knockdown ALP staining also indicated higher staining in overexpressed cells. Scale Bar = 200 µm. (**D**) Quantification of ALP and alizarin staining was performed using NIH image J, *** *p* < 0.001, *n* = 3.

**Figure 5 cells-10-03518-f005:**
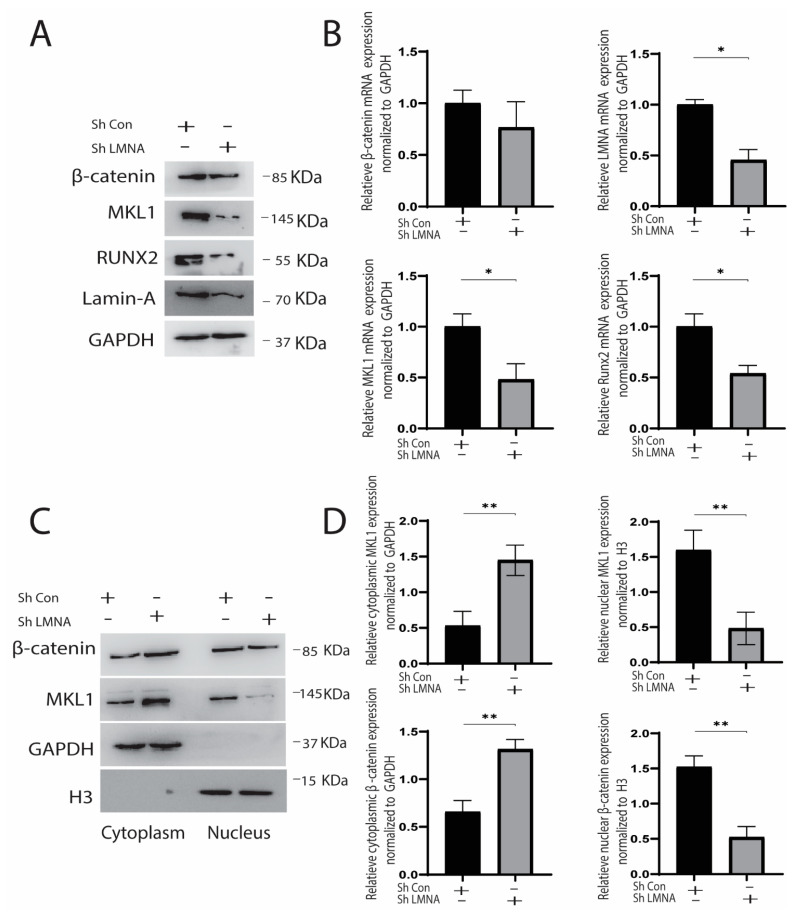
(**A**) LMNA knockdown inhibited mechanical strain-induced osteogenesis by inhibiting RUNX2 and decreasing MKL1 and β-catenin (Western blot analysis); (**B**) mRNA expression using RT-qPCR, * *p* < 0.05, *n* = 3; (**C**) MKL1 and β-catenin translocation in the nucleus was reduced by LMNA knockdown shown by protein expression using Western blot analysis; (**D**) Densitometric quantification of the Western blotting bands normalized to GAPDH for cytoplasmic protein and H3 for nuclear protein. ** *p* < 0.01, *n* = 3.

**Figure 6 cells-10-03518-f006:**
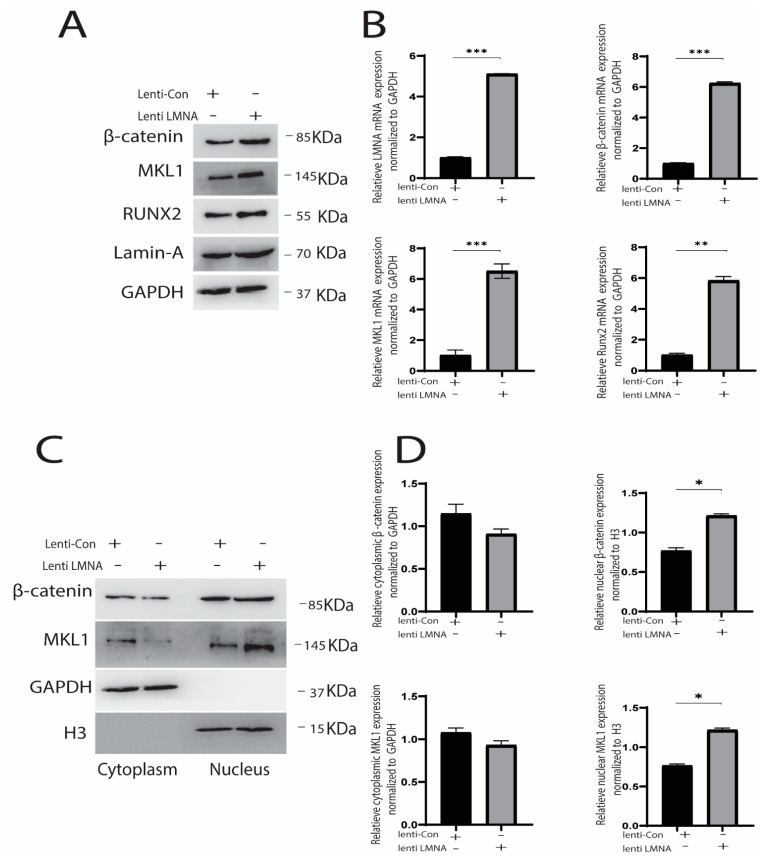
(**A**) LMNA overexpression increased mechanical strain-induced osteogenesis by increasing RUNX2, MKL1, and β-catenin (Western blot analysis); (**B**) mRNA expression using RT-qPCR, * *p* < 0.05, ** *p* < 0.01, ****p* < 0.001, *n* = 3; (**C**) MKL1 and β-catenin translocated in the nucleus by LMNA overexpression shown by protein expression using Western blot analysis; (**D**) Densitometric quantification of the Western blotting bands normalized to GAPDH for cytoplasmic protein and H3 for nuclear protein. * *p* < 0.05, *n* = 3.

**Figure 7 cells-10-03518-f007:**
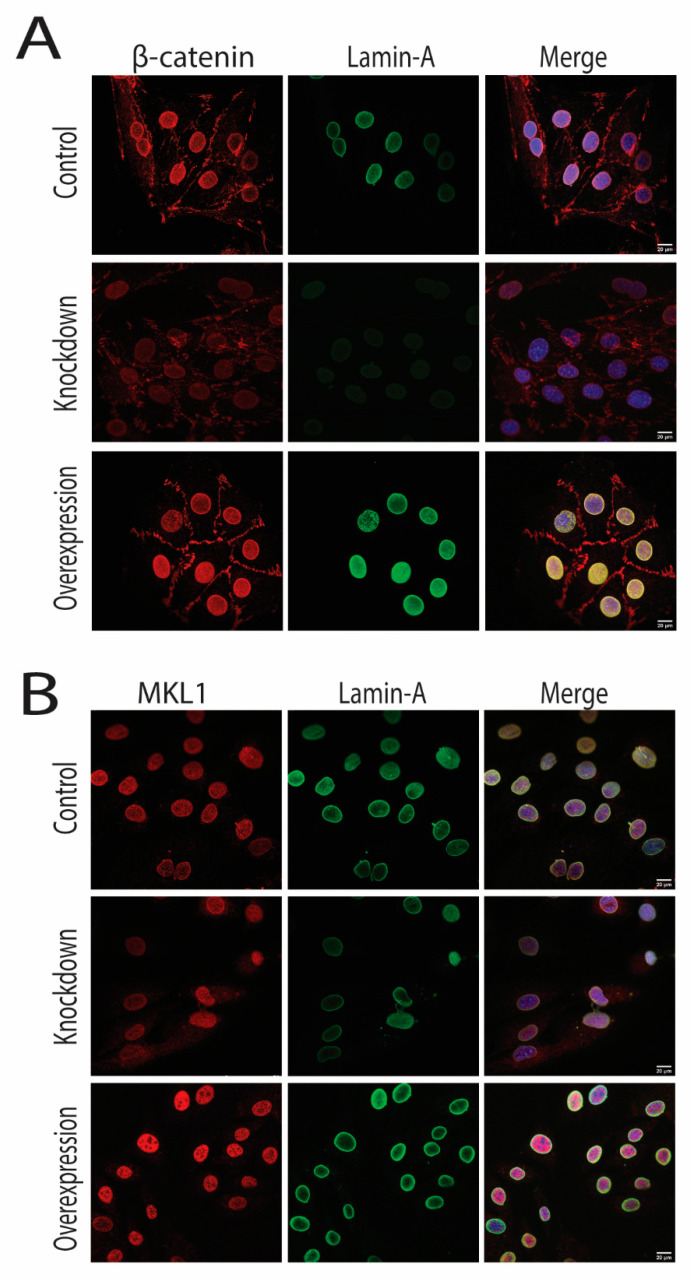
(**A**) Immunofluorescence results showed that LMNA overexpression and knockdown promoted an increased translocation of β-catenin in the nucleus and cytoplasm, respectively; (**B**) Similarly, MKL1 was also expressed more in the nucleus in overexpressed LMNA cells than knockdown LMNA cells. Scale bar = 20 µm.

**Table 1 cells-10-03518-t001:** Primers used in the qPCR gene.

Gene	Forward Primer (5→3)	Reverse Primer (3→5)
GAPDH	CAATGTGTCCGTCGTGGATCT	GTCCTCAGTGTAGCCCAAGATG
LMNA	CCTTCGCACCGCTCTCATCAAC	TCTTCTCCATCCTCGTCGTCATCC
RUNX2	TCCCGTCACCTCCATCCTCTTTC	GAATACGCATCACAACAGCCACAAG
ALP	CTTGGTGGTCACAGCAGTTGGTAG	CCAGGCGACAGGTGAAGAAACAG
MKL1	GTGCTGCGTCCTGCTGTCTAAG	GCTCCTCAATCTGCTTGTCCTTCTC
β-catenin	GCTGCTGTCCTATTCCGAATGTCTG	GGCACCAATGTCCAGTCCAAGATC

## Data Availability

All data generated or analysed during this study are included in this published article.

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
