# Peer review of "Lamin A/C-Dependent Translocation of Megakaryoblastic Leukemia-1 and β-Catenin in Cyclic Strain-Induced Osteogenesis"

_cells, 2021, doi:10.3390/cells10123518_

Round 1
Reviewer 1 Report
The manuscript entitled “Lamin A/C dependent translocation of megakaryoblastic leukemia-1 and β-catenin in cyclic strain-induced osteogenesis” described in a very clear manner the effects of LMNA in inducing osteogenesis during cyclic strain. The paper displayed several analysis, including RT-PCR, WB and IF to support the theory that translocation of MKL1 and β-catenin from cytoplasm to nucleus is dependent on LMNA espression.
Minor comments:
- Introduction: please add some references about the effects of the mechanical loading on bone differentiation, including for example the whole-body vibration that mimics the locomotion. For this reason you can cite these papers: (i) Prè D, Ceccarelli G, Visai L, Benedetti L, Imbriani M, Cusella De Angelis MG, Magenes G. High-Frequency Vibration Treatment of Human Bone Marrow Stromal Cells Increases Differentiation toward Bone Tissue. Bone Marrow Res. 2013;2013:803450. doi: 10.1155/2013/803450. Epub 2013 Mar 25. PMID: 23585968; PMCID: PMC3621160; (ii) Prè D, Ceccarelli G, Gastaldi G, Asti A, Saino E, Visai L, Benazzo F, Cusella De Angelis MG, Magenes G. The differentiation of human adipose-derived stem cells (hASCs) into osteoblasts is promoted by low amplitude, high frequency vibration treatment. Bone. 2011 Aug;49(2):295-303. doi: 10.1016/j.bone.2011.04.013. Epub 2011 Apr 30. PMID: 21550433. Cyclic strain and other mechanical stress like vibration could be similar applied on bone cells? Please specify.
- Materials and Methods: please add a picture of the mechanical strain used to apply stress to MC3T3-E1 cells in order better understand the bioreactor. Cells were cultivated in proliferative or osteogenic conditions? How are you sure that already at 7 days they have become osteoblasts?
- Results: Authors know the limitations of this study. The process of osteogenesis is very long, so it would be very useful to follow traslocation of MLK-1 and β-catenin induced by LMNA also at 20 or 30 days of culture. The effect is maintained? Is it lost? Increase? Is it possbile to add some results maybe at 15 or 20 days? Or maybe at 4/5 hours after stimulation? What are the expression of genes more osteoblast-like, as Osteocalcin or osteonectin? Please add these data if possible.
Author Response
We gratefully thank editors and all reviewers for spending their time to make their constructive comments. These suggestions have significantly improved the quality of the manuscript and indicated the direction in which we need to explore further. We have read and considered each suggestion and comment. Below the comments of the reviewers are response and our revisions are indicated.
Reviewer 1
The manuscript entitled “Lamin A/C dependent translocation of megakaryoblastic leukemia-1 and β-catenin in cyclic strain-induced osteogenesis” described in a very clear manner the effects of LMNA in inducing osteogenesis during cyclic strain. The paper displayed several analysis, including RT-PCR, WB and IF to support the theory that translocation of MKL1 and β-catenin from cytoplasm to nucleus is dependent on LMNA espression.
Minor comments:
- Introduction: please add some references about the effects of the mechanical loading on bone differentiation, including for example the whole-body vibration that mimics the locomotion. For this reason you can cite these papers: (i) Prè D, Ceccarelli G, Visai L, Benedetti L, Imbriani M, Cusella De Angelis MG, Magenes G. High-Frequency Vibration Treatment of Human Bone Marrow Stromal Cells Increases Differentiation toward Bone Tissue. Bone Marrow Res. 2013;2013:803450. doi: 10.1155/2013/803450. Epub 2013 Mar 25. PMID: 23585968; PMCID: PMC3621160; (ii) Prè D, Ceccarelli G, Gastaldi G, Asti A, Saino E, Visai L, Benazzo F, Cusella De Angelis MG, Magenes G. The differentiation of human adipose-derived stem cells (hASCs) into osteoblasts is promoted by low amplitude, high frequency vibration treatment. Bone. 2011 Aug;49(2):295-303. doi: 10.1016/j.bone.2011.04.013. Epub 2011 Apr 30. PMID: 21550433. Cyclic strain and other mechanical stress like vibration could be similar applied on bone cells? Please specify.
Author response:
Thank you very much for this suggestion. We have already added these two references in introduction. Besides. To make the reader understand, we also added several other references where the cyclic strain was applied on different cell types and enhanced osteogenic differentiation.
One study indicated that high-frequency vibration augmented the osteogenic differentiation in bone marrow stem cells (BMSCs)[17]. Also, low amplitude high-frequency was reported to increase the bone formation process in the initial stages[18]. Similarly, the cyclic strain also increased extracellular matrix mineralization through extracellular signal-regulated kinase (ERK1/2) human mesenchymal stem cells [19]. Multiple other studies also indicated the increased osteogenic differentiation in human-derived adipose stem cells (hASCs)[20, 21], BMSC [22-24], human intraoral mesenchymal stem and progenitor cells [25], Human embryonic stem cells (hESCs) [26], human fibroblast [27] and MC3T3-E1 cells [28, 29]under the influence of cyclic tensile stress.
Comment:
Materials and Methods: please add a picture of the mechanical strain used to apply stress to MC3T3-E1 cells in order better understand the bioreactor. Cells were cultivated in proliferative or osteogenic conditions?
Author response:
Thank you very much for the suggestion. We have already added a picture of FlexCells used for cyclic stress on the cells. Cells were cultured in a growth medium throughout our study, and we have followed the following articles for that.
- Singh, S. P., Chang, E. I., Gossain, A. K., Mehara, B. J., Galiano, R. D., Jensen, J., ... & Saadeh, P. B. (2007). Cyclic mechanical strain increases production of regulators of bone healing in cultured murine osteoblasts. Journal of the American College of Surgeons, 204(3), 426-434.
- Luo, Y., Ge, R., Wu, H., Ding, X., Song, H., Ji, H., ... & Du, H. (2019). The osteogenic differentiation of human adipose-derived stem cells is regulated through the let-7i-3p/LEF1/β-catenin axis under cyclic strain. Stem cell research & therapy, 10(1), 1-19.
- Charoenpanich, A., Wall, M. E., Tucker, C. J., Andrews, D. M., Lalush, D. S., Dirschl, D. R., & Loboa, E. G. (2014). Cyclic tensile strain enhances osteogenesis and angiogenesis in mesenchymal stem cells from osteoporotic donors. Tissue Engineering Part A, 20(1-2), 67-78.
- Nam, H. Y., Pingguan-Murphy, B., Abbas, A. A., Merican, A. M., & Kamarul, T. (2019). Uniaxial cyclic tensile stretching at 8% strain exclusively promotes tenogenic differentiation of human bone marrow-derived mesenchymal stromal cells. Stem cells international, 2019.
- Peng, Y., Qu, R., Feng, Y., Huang, X., Yang, Y., Fan, T., ... & Ouyang, J. (2021). Regulation of the integrin αVβ3-actin filaments axis in early osteogenesis of human fibroblasts under cyclic tensile stress. Stem Cell Research & Therapy, 12(1), 1-12.
Comment:
How are you sure that already at 7 days they have become osteoblasts?
Author response:
MC3T3-E1 are murine osteoblast cells line which is exclusively used to study osteogenic differentiation. In fact, pre-osteoblast cells take a long time to become osteoblast and osteocyte; however, our study focused only on the early stages of osteogenesis using cyclic strain. This is the reason we only studied 1, 4, and 7 day in our project. To ensure the early stage of osteogenic differentiation, we used ALP and RUNX2 markers along with ALP staining. Our result indicated that osteogenic differentiation was successfully started, and two signalling molecules of Wnt/ β-catenin and MKL1/SRF, essential for osteogenesis, were translocated in the nucleus over time. We also added several references in the introduction as per your suggestion to support the idea that cyclic strain encouraged osteogenic differentiation.
Comment:
Results: Authors know the limitations of this study. The process of osteogenesis is very long, so it would be very useful to follow traslocation of MLK-1 and β-catenin induced by LMNA also at 20 or 30 days of culture. The effect is maintained? Is it lost? Increase? Is it possbile to add some results maybe at 15 or 20 days? Or maybe at 4/5 hours after stimulation? What are the expression of genes more osteoblast-like, as Osteocalcin or osteonectin? Please add these data if possible.
Author response:
We highly appreciate this comment. Our present study mainly focused on the early stages of the strain-induced osteogenic process wherein signalling molecules MKL1 and β-catenin were dependent on LMNA immediately after the stimulation. Infect, as you stated, we have mentioned in our study that there were a few limitations in our research that will be addressed in our future study. However, we have done some preliminary western blot experiments after 30 minutes, 2 , 4 and 8 hours of stimulation on day 7. We found that MKL1 showed some cytoplasmic translocation while retaining their expression at the nucleus. As per your recommendation, we have added results given below as supplementary files.
Reviewer 2 Report
This manuscript studies the relationship between lamins and molecules involved in osteogenesis induced by mechanical stress..
The design of the study is appropriate. However, a number of questions and concerns need to be revised prior publication:
Specific comments for revision:
In the second paragraph of the Introduction section, the last part has to be rewrited as it is not clear what the authors mean with: “One possible hypothesis is that Lamin mutation cause disruption in cell signaling and gene expression, which is also supported by a few studies such as MKL1/SRF (megakaryoblastic leukemia 1/ Serum response factor) signaling.
The authors state that the role of LMNA in osteogenes has “recently” gained interest, but the references are from 10 years ago, so it is not so recently.
In figure 1 legend, is A protein expression? It is not clear.
In the text and figure legends some B-catenin are typed instead of β-catenin. Please revise it.
In figure 5 A, the expression of lamin-A is not completely abrogated so the study should be improved by another knockdown technique.
In Materials and Methods, the secondary antibody used for immunofluorescence is not explained. The lysis buffer used in western blotting is not described and the species of the anti-ALP antibody is not defined.
Author Response
We gratefully thank editors and all reviewers for spending their time to make their constructive comments. These suggestions have significantly improved the quality of the manuscript and indicated the direction in which we need to explore further. We have read and considered each suggestion and comment. Below the comments of the reviewers are response and our revisions are indicated.
Reviewer 2
This manuscript studies the relationship between lamins and molecules involved in osteogenesis induced by mechanical stress..
The design of the study is appropriate. However, a number of questions and concerns need to be revised prior publication:
Specific comments for revision:
Comment:
In the second paragraph of the Introduction section, the last part has to be rewrited as it is not clear what the authors mean with: “One possible hypothesis is that Lamin mutation cause disruption in cell signaling and gene expression, which is also supported by a few studies such as MKL1/SRF (megakaryoblastic leukemia 1/ Serum response factor) signaling.
Author response
Thank you very much for writing this comment. We already revised to to clarify what we meant in that line.
One hypothesis is that Lamin mutation cause disruption in cell signaling and gene expression, which is also supported by a study wherein, fibroblast cells showed an interrupted MKL1/SRF signaling in LMNA depleted cells.[10].
Comment:
The authors state that the role of LMNA in osteogenes has “recently” gained interest, but the references are from 10 years ago, so it is not so recently.
Author response
Thank you very much for highlighing this issue. we deleted this word in text and we have updated the following references in that line.
- Alcorta-Sevillano, N., Macías, I., Rodríguez, C. I., & Infante, A. (2020). Crucial role of Lamin A/C in the migration and differentiation of MSCs in bone. Cells, 9(6), 1330.
- Malashicheva, A., & Perepelina, K. (2021). Diversity of Nuclear Lamin A/C Action as a Key to Tissue-Specific Regulation of Cellular Identity in Health and Disease. Frontiers in Cell and Developmental Biology, 2834.
Comment:
In figure 1 legend, is A protein expression? It is not clear.
In the text and figure legends some B-catenin are typed instead of β-catenin. Please revise it.
Author response
Thank you very much for suggesting me improvement. We modified the legend of figure 1 “In figure 1, (A) is protein expression. Words B-catenin in the manuscript have been replaced with β-catenin”.
Comment:
In figure 5 A, the expression of lamin-A is not completely abrogated so the study should be improved by another knockdown technique.
Author response
Thank you very much for raising this question. In fact, we used Sh plasmid Lamin-A knockdown, so we were not expecting complete abrogation because complete abrogation/ knockout LMNA might cause nuclear breakage which may lead to apoptosis of cells under strain (references are given below). This death and nuclear breakage might have caused hindrance in our experiments because all our experiments were based on cyclic strain and translocation of molecules. This is the reason we use Sh plasmid LMNA to partially knockdown instead of complete knockout.
- Suchitra Chandar, Li Sze Yeo, Christiana Leimena, et al. (2010). Effects of mechanical stress and carvedilol in lamin A/C–deficient dilated cardiomyopathy. Circulation research, 106(3), 573-582.
- Vesna Nikolova, Christiana Leimena, Aisling C. McMahon, et al. (2004). Defects in nuclear structure and function promote dilated cardiomyopathy in lamin A/C–deficient mice. The Journal of clinical investigation, 113(3), 357-369.
- Olga Moiseeva, Ve´ronique Bourdeau, Mathieu Vernier, Marie-Christine Dabauvalle, Gerardo Ferbeyre (2011). Retinoblastoma‐independent regulation of cell proliferation and senescence by the p53–p21 axis in lamin A/C‐depleted cells. Aging cell, 10(5), 789-797.
- Jos L V Broers, Emiel A G Peeters, Helma J H Kuijpers, et al. (2004). Decreased mechanical stiffness in LMNA−/− cells is caused by defective nucleo-cytoskeletal integrity: implications for the development of laminopathies. Human molecular genetics, 13(21), 2567-2580.
Comment:
In Materials and Methods, the secondary antibody used for immunofluorescence is not explained. The lysis buffer used in western blotting is not described and the species of the anti-ALP antibody is not defined.
Author response
Thank you very much for the comment. I have added the information missed by us previously.
The secondary antibodies have been written and added in the manuscript
Actually, we purchased the protein extraction kit, and lysis buffer were prepared according to given instruction by the company. Besides, species of antibody has been updated as well.
“After harvesting and washing the cells, the protein was extracted by preparing immunoprecipitation lysis buffer using the instruction given in whole protein extraction kit (Key GEN BioTECH, Nanjing, China)”
Reviewer 3 Report
The authors studied the role of MKL1 and beta-catenin in mechanical strain sensing of MSCs undergoing osteogenic differentiation with a knock out/overexpression setting. Interestingly, osteogenesis by MKL1 was regulated by lamin A/C.
General question: why does this solely stretching setting trigger exclusively osteogenesis and not rather simultaneously stimulate ligamento-/tenogenesis? Were tendon markers included in the analyses? Osteogenesis would also require pressure load in combination with stretch.
On which consideration was the training profil of the cells selected?
Does the shape of the cell nuclei change (obviously not)? Lamins might stabilize cell nucleis envelope.
Flex cell culture plates: this model does not garantee a strictly uniaxial strain but rather a inhomogenous strain profil.
minor
"Knockdown" was sometimes written in capital letters or even not. (e.g. 2.3.), please adapt
page 2 last paragraph of introduction: write "ß-catenin" as before.
Figures: 1A/2A-B: westernblot? please indicate that the molecular weights are kd (?) e.g. H3 (reference protein?), please explain generally the abbreviations used in the figures in the legend. Fig. 5C: why is H3 absent in the first two lines? Legend of Figure 5/6 (Legends) add missing blanks, Fig. 5 (Legend) write ßcatenin with hyphen.
page 11: "in overexpressed cells" explain what the cells overexpress "than IN knock down"... last paragraph: bring the citations in the same bracket. Add lacking blanks (e.g. before citation). "knock down result", write "results".
4.7. where is H3 mentioned? were nuclear and cytoplasmic extracts prepared (figure 2)?
figure 3: indicate generally the stretch direction. In which direction do the cells align their cytoskeleton in response to strain? I would not use the term "loading" since no pressure but stretch was applied.
alizarin red and ALP stain should be quantified
Author Response
We gratefully thank editors and all reviewers for spending their time to make their constructive comments. These suggestions have significantly improved the quality of the manuscript and indicated the direction in which we need to explore further. We have read and considered each suggestion and comment. Below the comments of the reviewers are response and our revisions are indicated.
Reviewer 3
The authors studied the role of MKL1 and beta-catenin in mechanical strain sensing of MSCs undergoing osteogenic differentiation with a knock out/overexpression setting. Interestingly, osteogenesis by MKL1 was regulated by lamin A/C.
Comment:
General question: why does this solely stretching setting trigger exclusively osteogenesis and not rather simultaneously stimulate ligamento-/tenogenesis? Were tendon markers included in the analyses? Osteogenesis would also require pressure load in combination with stretch.
Author response
Thank you for raising this interesting question. There are few reasons why we think that cyclic strain increase osteogenesis.
Firstly, we aimed to focus on the osteogenic process solely. That is why we use MC3T3-E1 cells, a murine osteoblast cell line, exclusively used to study osteogenic differentiation. If we had used mesenchymal stem cells, we would expect some chondrogenesis, but the use of osteoblast cell line elaminate the chances of differentiation into chondrogenesis.
Secondly, we use RUNX2 and ALP as osteogenic markers and ALP staining to confirm the initiation of the osteogenic process.
Finally, various studies on the internet have observed the cyclic strain’s effect on different cells, including osteoblast, bone marrow stem cells, and adipose-derived stem cells. All studies suggested that cyclic strain encouraged osteogenesis in these cells. A few of the references are below:
- Simmons, C.A., et al., Cyclic strain enhances matrix mineralization by adult human mesenchymal stem cells via the extracellular signal-regulated kinase (ERK1/2) signaling pathway. Journal of biomechanics, 2003. 36(8): p. 1087-1096.
- Luo, Y., et al., The osteogenic differentiation of human adipose-derived stem cells is regulated through the let-7i-3p/LEF1/β-catenin axis under cyclic strain. Stem cell research & therapy, 2019. 10(1): p. 1-19.
- Charoenpanich, A., et al., Microarray analysis of human adipose-derived stem cells in three-dimensional collagen culture: osteogenesis inhibits bone morphogenic protein and Wnt signaling pathways, and cyclic tensile strain causes upregulation of proinflammatory cytokine regulators and angiogenic factors. Tissue Engineering Part A, 2011. 17(21-22): p. 2615-2627.
- Charoenpanich, A., et al., Cyclic tensile strain enhances osteogenesis and angiogenesis in mesenchymal stem cells from osteoporotic donors. Tissue Eng Part A, 2014. 20(1-2): p. 67-78.
- Qi, M.c., et al., expression of bone‐related genes in bone marrow MSCs after cyclic mechanical strain: implications for distraction osteogenesis. International journal of oral science, 2009. 1(3): p. 143-150.
- Sumanasinghe, R.D., S.H. Bernacki, and E.G. Loboa, Osteogenic differentiation of human mesenchymal stem cells in collagen matrices: effect of uniaxial cyclic tensile strain on bone morphogenetic protein (BMP-2) mRNA expression. Tissue engineering, 2006. 12(12): p. 3459-3465.
Comment:
On which consideration was the training profil of the cells selected?
Author response
Thank you for this comment. Infect, we aimed to solely focus on the signalling molecules involved in the osteogenic process. That is why we use MC3T3-E1 cells, a murine osteoblast cell line, which is widely used to study osteogenic differentiation.
Comment:
Does the shape of the cell nuclei change (obviously not)? Lamins might stabilize cell nucleis envelope.
Author response
Thank you for asking another interesting question. We have not observed any change in the size of nucleus. Complete knockout cells might show the change in the nucleus morphology but in our study knockdown did not have any change in the size of the nucleus. knockout LMNA might cause nuclear breakage which may lead to apoptosis of cells under strain (references are given below). This is one of the reasons we did not use complete knockout in our experiments.
- Suchitra Chandar, Li Sze Yeo, Christiana Leimena, et al. (2010). Effects of mechanical stress and carvedilol in lamin A/C–deficient dilated cardiomyopathy. Circulation research, 106(3), 573-582.
- Vesna Nikolova, Christiana Leimena, Aisling C. McMahon, et al. (2004). Defects in nuclear structure and function promote dilated cardiomyopathy in lamin A/C–deficient mice. The Journal of clinical investigation, 113(3), 357-369.
- Olga Moiseeva, Ve´ronique Bourdeau, Mathieu Vernier, Marie-Christine Dabauvalle, Gerardo Ferbeyre (2011). Retinoblastoma‐independent regulation of cell proliferation and senescence by the p53–p21 axis in lamin A/C‐depleted cells. Aging cell, 10(5), 789-797.
- Jos L V Broers, Emiel A G Peeters, Helma J H Kuijpers, et al. (2004). Decreased mechanical stiffness in LMNA−/− cells is caused by defective nucleo-cytoskeletal integrity: implications for the development of laminopathies. Human molecular genetics, 13(21), 2567-2580.
Comment:
Flex cell culture plates: this model does not garantee a strictly uniaxial strain but rather a inhomogenous strain profil.
Author response
Thank you for adding this point. We agree that this model provide inhomogenious strain however, the information given by the FlexCells is that it provide uniaxial strain. We have not described the uniaxial strain in the paper.
Minor Comment:
"Knockdown" was sometimes written in capital letters or even not. (e.g. 2.3.), please adapt
page 2 last paragraph of introduction: write "ß-catenin" as before.
Author response
Thank you for correcting us. We have already made correction of these in following areas.
“We then knocked down and overexpressed LMNA to investigate its effect on the translocation of signaling molecules. Sh plasmid and lentivirus were used for the knockdown and the overexpression of LMNA, respectively. Fluorescence microscopy was used to observe Green fluorescent protein (GFP), which showed that transfection efficiency was above 80 % (Fig. 4 A). Both knockdown and overexpression LMNA were then confirmed using Western Blot, RT-qPCR, and immunofluorescence (Fig. 4 B). In-terestingly, Phalloidin staining revealed more distorted actin filaments in the knock-down LMNA cells over the nucleus, whereas abundant stress fibers were seen in con-trol and LMNA overexpressed cell (Fig. 4 B).”
In introduction “Furthermore, it is also reported that both MKL1 and β-catenin are mechano-sensitive signaling molecules that respond to the mechanical strain”
In 2.2 “Similarly, β-catenin also decreased gradually over time in the cytoplasm, and the accumulation of β-catenin increased significantly in the nucleus by following the same trend”
Comment:
Figures: 1A/2A-B: westernblot? please indicate that the molecular weights are kd (?) e.g. H3 (reference protein?), please explain generally the abbreviations used in the figures in the legend. Fig. 5C: why is H3 absent in the first two lines? Legend of Figure 5/6 (Legends) add missing blanks, Fig. 5 (Legend) write ßcatenin with hyphen.
Author response
Thank you for correcting us. We have already made the correction and updated figures by putting KDa in western blot pictures. Correction in the legends are as follows
Figure 1. (A)The FX-5000 system (Flexcell International), the direction of cyclic strain, and the orientation of actin cell cytoskeleton parallel to the direction of strain. (B) The Protein expression of β-catenin, MKL1 (megakaryoblastic leukaemia-1), ALP (Alkaline phosphatase), RUNX2 (Runt-related transcription factor 2) and LMNA (Lamin A/C) in MC3T3-E1 cells under cyclic strain; β-catenin, MKL1, ALP, RUNX2, LMNA protein expression increased gradually and peaked at 7th day(western blot analysis); (C) The relative mRNA upregulated expression of β-catenin, MKL1, ALP, RUNX2 and LMNA (Lamin A/C) in MC3T3-E1 cells under cyclic strain; (D)Quantification of ALP staining was performed using NIH image J, * P < 0.05, ** P < 0.01, *** P < 0.001, n = 3; ALP staining on day 0, 4 and 7 showed increased staining at 7th day as compared to day 0; (E)The immunofluorescence expression of Lamin-A in MC3T3-E1 cells under cyclic strain and without strain. Scale bar = 20 µm
Figure 2. (A) The Protein expression of MKL1 and β-catenin decreased significantly in the cytoplasm on days 4 and 7(western blot analysis); (B) The Protein expression of MKL1 and β-catenin increased significantly in the nucleus on day 7(western blot analysis); (C) The graph shows that MKL1 increased on the first day in the cytoplasm and gradually decreased; (D) The graph shows that MKL1 and β-catenin increased gradually in the nucleus and peaked on the 7th day. Densitometric quantification of the Western blotting bands normalized to Glyceraldehyde 3-phosphate dehydrogenase (GAPDH) for cytoplasmic protein and Histone (H3) for nuclear protein * P < 0.05, ** P < 0.01, *** P < 0.001, n = 3.
Figure 3. The immunofluorescence expression of β-catenin and MKL1 in MC3T3-E1 cells after a stretch without stretch wherein stress fibres aligned parallel to the direction of strain. (A) β-catenin expression was cytoplasmic and nuclear in control and 1st, but it indicates nuclear accumulation on the 4th and 7th day (B). The expression of MKL1 was more cytoplasmic on the control and 1st day, but it showed higher nuclear translocation on the 4th and 7th day. Scale bar = 20 µm
Comment:
page 11: "in overexpressed cells" explain what the cells overexpress "than IN knock down"... last paragraph: bring the citations in the same bracket. Add lacking blanks (e.g. before citation). "knock down result", write "results".
Author response
Thank you for highlighting these mistakes. We have already corrected following in our paper. Citations were added together and grammatically incorrect sentence was corrected. We have corrected the sentense and made it past as last lines were describing the past tense.
“In our study, Alizarin staining was more evident in LMNA overexpressed cells, indicating more osteogenesis in LMNA overexpressed cells than LMNA knockdown and control. LMNA knockdown decreased the MKL1, RUNX2 expression, and its overexpression increased MKL1, β-catenin, and RUNX2 expression”
“The evidence also supports that in LMNA deficient fibroblast cells, most of MKL1 were translocated in the cytoplasm leaving behind very little in the nucleus [10, 62]”
“Moreover, LMNA overexpression resulted in a higher amount of actin fibers and knockdown resulted in actin degradation, especially around the nucleus.”
Comment:
4.7. where is H3 mentioned? were nuclear and cytoplasmic extracts prepared (figure 2)?
Author response
Thank you for the comment. We have updated the legend and mentioned H3. the updated legend is written here below. Cytoplasmic and nuclear protein was extracted using Cytoplasmic Protein Extraction Kit (Keygen Biotech, Nanjing, China) as mentioned in the material and method.
“Figure 2. (A) The Protein expression of MKL1 and β-catenin decreased significantly in the cytoplasm on days 4 and 7(western blot analysis); (B) The Protein expression of MKL1 and β-catenin increased significantly in the nucleus on day 7(western blot analysis); (C) The graph shows that MKL1 increased on the first day in the cytoplasm and gradually de-creased; (D) The graph shows that MKL1 and β-catenin increased gradually in the nucleus and peaked on the 7th day. Densitometric quantification of the Western blotting bands normalized to Glyceraldehyde 3-phosphate dehydrogenase (GAPDH) for cytoplasmic protein and Histone (H3) for nuclear protein * P < 0.05, ** P < 0.01, *** P < 0.001, n = 3.”
Comment:
figure 3: indicate generally the stretch direction. In which direction do the cells align their cytoskeleton in response to strain? I would not use the term "loading" since no pressure but stretch was applied.
alizarin red and ALP stain should be quantified
Author response
Thank you very much for guiding us. We have updated figure 1 where we have added the direction of cyclic strain on the schematic diagram to show the general cyclic stretch and stress fibres orientation (Fig, 1A, B). We changed the general term “stretch” instead of loading to make it easier for the reader to understand. Also, Alizarin and ALP staining has been graphically expressed after quantification.
Reviewer 4 Report
Asmat et al. showed increased nuclear translocation of b-catenin, and MKL-1 after cyclic stain application on murine osteoblast precursor cell line (MC3T3). Moreover, knockdown of lamin A/C reduced this nuclear translocation, and overexpression of lamin A/C enhanced this nuclear translocation of b-catenin, and MKL-1. These results are interesting. I have some comments to the authors:
Introduction:
The authors described role of lamin in detail. Mechanical force is transferred from the cytoskeleton to the nucleoskeleton through tethering of actin to Linker of Nucleus and Cytoskeleton (LINC) complexes. They need to describe more about Wnt/b-catenin and MK1/SRF (when you first mention the word, write the full name “serum response factor”) and their roles in actin and LINC complex in this section. The following are to publications related to the role of MKL1 and /b-catenin on actin and LINC.
Mkl1-dependent gene activation is sufficient to induce actin cap assembly.
Thakar K, Carroll CW.
Small GTPases. 2019 Nov;10(6):433-440. doi: 10.1080/21541248.2017.1328303. Epub 2017 Jul 7.PMID: 28586283
Gene regulation through dynamic actin control of nuclear structure.
Sankaran J, Uzer G, van Wijnen AJ, Rubin J.Exp Biol Med (Maywood). 2019 Nov;244(15):1345-1353. doi: 10.1177/1535370219850079. Epub 2019 May 13.
Results:
- The authors firstly described RUNX2 and ALP in Fig 1, then Lamin A/C, and finally MKL1 and b-catenin. However, Fig. 1A listed MKL1 and b-catenin ahead of ALP, RUNX2, and Lamin A/C. Fig. 1B also listed Lamin A, b-catenin first, but Runx2, ALP last. I recommend to rearrange the Fig. 1A, Fig 1B sequence (first RUNX2, ALP, then, Lamin A/C, finally, MKL1 and b-catenin).
- 4C Alizarin staining was weak. This picture does not show red color as stained by alizarin red S, but shows blue color. Is this double staining of ALP with Alizarin red S? I suggest to show higher magnification of Alizarin red S staining.
- 5C and Fig. 6C should label “cytoplasm” and “nuclear” in the figures.
- 7 only shows nuclear expression of MKL1 and b-catenin. However, the figure legend described that (A) translocation of b-catenin in the nucleus and cytoplasm. (B) MKL1 was also expressed more in the cytoplasm in overexpressed LMNA cells. In overexpressed cells, the MKL1 only expressed in the nucleus. I did not see the cytoplasmic expression of MKL1.
Author Response
We gratefully thank editors and all reviewers for spending their time to make their constructive comments. These suggestions have significantly improved the quality of the manuscript and indicated the direction in which we need to explore further. We have read and considered each suggestion and comment. Below the comments of the reviewers are response and our revisions are indicated.
Reviewer4
Asmat et al. showed increased nuclear translocation of b-catenin, and MKL-1 after cyclic stain application on murine osteoblast precursor cell line (MC3T3). Moreover, knockdown of lamin A/C reduced this nuclear translocation, and overexpression of lamin A/C enhanced this nuclear translocation of b-catenin, and MKL-1. These results are interesting. I have some comments to the authors:
Introduction:
The authors described role of lamin in detail. Mechanical force is transferred from the cytoskeleton to the nucleoskeleton through tethering of actin to Linker of Nucleus and Cytoskeleton (LINC) complexes. They need to describe more about Wnt/b-catenin and MK1/SRF (when you first mention the word, write the full name “serum response factor”) and their roles in actin and LINC complex in this section. The following are to publications related to the role of MKL1 and /b-catenin on actin and LINC.
Mkl1-dependent gene activation is sufficient to induce actin cap assembly.
Thakar K, Carroll CW. Small GTPases. 2019 Nov;10(6):433-440. doi: 10.1080/21541248.2017.1328303. Epub 2017 Jul 7.PMID: 28586283
Gene regulation through dynamic actin control of nuclear structure. Sankaran J, Uzer G, van Wijnen AJ, Rubin J.Exp Biol Med (Maywood). 2019 Nov;244(15):1345-1353. doi: 10.1177/1535370219850079. Epub 2019 May 13
Author response
Thank you for suggesting this addition. I explained briefly about Wnt/ β -catenin and MKL1/SRF in the introduction as per your suggestion.
“Previous reports have established that Wnt/ β -catenin and MKL1/SRF pathways are regulatory factors for osteogenesis[37-39]. Being the essential part of adherent junction, β -catenin regulates cell adhesion and activates Wnt cell signaling. Nuclear translocation of β -catenin causes the activation of specific genes through transcription factors mediating cellular development [40, 41]. Activation of Wnt/ β -catenin signaling results in the mouse mesenchymal stem cells differentiation[42]. Similarly, translocation of MKL1 is mainly regulated by actin cytoskeleton, which is coupled with Lamin A/C through Linker of Nucleoskeleton and Cytoskeleton (LINC) complexes[43]. The state of actin depolymerization results in higher G-actin monomers, which binds with MKL1 and help its translocation in the nucleus for the target genes[44]. MKL1/SRF activation also results in stress fiber essential for mechanotransdcution[43].”
Comment:
Results:
The authors firstly described RUNX2 and ALP in Fig 1, then Lamin A/C, and finally MKL1 and b-catenin. However, Fig. 1A listed MKL1 and b-catenin ahead of ALP, RUNX2, and Lamin A/C. Fig. 1B also listed Lamin A, b-catenin first, but Runx2, ALP last. I recommend to rearrange the Fig. 1A, Fig 1B sequence (first RUNX2, ALP, then, Lamin A/C, finally, MKL1 and b-catenin).
Author response
Thank you very much for guiding me to rectify our manuscript. We have updated the figure1 as per your suggestion.
Comment:
4C Alizarin staining was weak. This picture does not show red color as stained by alizarin red S, but shows blue color. Is this double staining of ALP with Alizarin red S? I suggest to show higher magnification of Alizarin red S staining.
Author response
We are thankful for the comment. We showed higher magnification of ALP staining and Alizarin staining as per your suggestion.
Comment:
5C and Fig. 6C should label “cytoplasm” and “nuclear” in the figures.
7 only shows nuclear expression of MKL1 and b-catenin. However, the figure legend described that (A) translocation of b-catenin in the nucleus and cytoplasm. (B) MKL1 was also expressed more in the cytoplasm in overexpressed LMNA cells. In overexpressed cells, the MKL1 only expressed in the nucleus. I did not see the cytoplasmic expression of MKL1
Author response
We highly appreciate your suggestions. We have added cytoplasm and nucleus in figures 5C and 6C to avoid confusion for the readers. Moreover, there was a mistake in the legend of Figure 7 B where the word should be “nuclear” rather than cytoplasmic, so we also corrected that. Following are the updates in the figures and legend.
“Figure 7. (A) Immunofluorescence result showed that LMNA overexpression and knockdown promoted an increased translocation of β-catenin in the nucleus and cytoplasm, respectively; (B) Similarly, MKL1 was also expressed more in the nucleus in overexpressed LMNA cells than knockdown LMNA cells. Scale bar = 20 µm.”
Round 2
Reviewer 1 Report
Dear authors,
thank you very much for answering my questions. The paper is now ready for publication.
Author Response
Thanks.
Reviewer 2 Report
The authors have addressed all my comments.
Author Response
Thanks.
Reviewer 3 Report
On which consideration was the training profil of the cells selected? This question is not answered.
legend of figure 4: " were performed using NIH image J" write "was"
concerning figure 3: "wherein stress fibres aligned parallel to the direction of strain." we know from our and other experiments that cells and stress fibers align rather in an angle to the stretch direction. This should be considered.
Page 7, second line: "significantly high staining in overexpressed LMNA (OE) cells as compared" write "higher"
Some issues should be adressed during proof reading: "Lamin" was corrected in the abstract ("lamin"), but it is also still written in other sections of the manuscript starting with capital letter. Page 3 second line: there is a surplus blank within the brackets. Legend Figure 2: "The Protein" protein should not be written in capital letters. There are a couple of lacking blanks and minor grammar errors throughout the whole manuscript.
Author Response
Comment
On which consideration was the training profil of the cells selected? This question is not answered.
Author response:
Thank you very much. We are sorry that we could not answer it properly.
First, our focus was the osteogenic process and signalling molecules associated with it in this study. So, we chose MC3T3 cells because they are used to study osteogenic differentiation exclusively. If we had chosen MSCs cells or fibroblast cells, we would have observed chondrogenesis or adipogenesis during the process. To eliminate the chances of chondrogenesis and adipogenesis in the study, we picked MC3T3 cells, which can only go for osteogenic differentiation under optimum conditions. Furthermore, we have gone through several studies on MC3T3 available on PubMed. Most studies them for osteogenic process and signaling pathways in different circumstances such as on different biomaterials, scaffolds, nanomaterial, etc.
Secondly, mechanotransduction is important for osteogenesis, so we wanted to apply strain on different time periods, specifically on osteoblast to see how signalling molecules are regulated and translocated under strain. We searched on google scholar and Pubmed and found many studies have already applied mechanical strain on MC3T3 cells so we chose MC3T3 cells for our study. Some of the references are given below:
- Rath, B., Nam, J., Knobloch, T. J., Lannutti, J. J., & Agarwal, S. (2008). Compressive forces induce osteogenic gene expression in calvarial osteoblasts. Journal of biomechanics, 41(5), 1095-1103.
- Kang, K. S., Lee, S. J., Lee, H., Moon, W., & Cho, D. W. (2011). Effects of combined mechanical stimulation on the proliferation and differentiation of pre-osteoblasts. Experimental & molecular medicine, 43(6), 367-373.
- Cora-Cruz, J. J., Diffoot-Carlo, N., & Sundaram, P. A. (2016). Vinculin expression in MC3T3-E1 cells in response to mechanical stimulus. Data in brief, 6, 94-100.
Finally, and most importantly, MC3T3-E1 has a similar growth rate to the human osteoblast due to the similarity in the molecular regulation of gene expression, such as in C/EBPβ - P1 promoter - Runx2/p57 axis. Moreover, proliferation and differentiation start early in the osteogenic process, and mineralization also starts early in MC3T3 cells. They are very useful at the early stages of differentiation, assessing the effect of the external intervention on the cells during the process (Czekanska et al., 2012).
- Czekanska, E. M., Stoddart, M. J., Richards, R. G., & Hayes, J. S. (2012). In search of an osteoblast cell model for in vitro research. Eur Cell Mater, 24(4), 1-17.
Ans:
Comment
legend of figure 4: " were performed using NIH image J" write "was"
Author response:
Thank you for guiding us again. This grammatical mistake has been corrected.
Figure 4. (A) MC3T3-E1 fluorescence intensity five days after virus infection. Scale bar = 200 µm; (B) The expression of knockdown (kd) and overexpression (OE) LMNA was detected by immunofluorescence. Scale Bar = 20 µm. Zoom in pictures showed that knockdown LMNA lacked actin fibers while overexpression showed intense phalloidin staining; (C) Alizarin staining showed that overexpression indicated more mineralization than control (CTL), and knockdown ALP staining also indicated higher staining in overexpressed cells Scale Bar = 200 µm; (D) Quantification of ALP and alizarin staining was performed using NIH image J, * p < 0.05, ** p < 0.01, *** p < 0.001, n = 3
Comment
concerning figure 3: "wherein stress fibres aligned parallel to the direction of strain." we know from our and other experiments that cells and stress fibers align rather in an angle to the stretch direction. This should be considered.
Author response:
Thank you for providing us insight about the direction of mechanical strain.
We have already added your suggestion and corrected it, and we followed the following paper, which mentioned the stress fiber was in an angle to the direction of strain.
Figure 3. The immunofluorescence expression of β-catenin and MKL1 in MC3T3-E1 cells after a stretch and without stretch wherein stress fibers aligned in an angle to the direction of strain. (A) β-catenin expression was cytoplasmic and nuclear in control and day 1st, but it indicates nuclear accumulation on the 4th and 7th day (B). The expression of MKL1 was more cytoplasmic on the control and 1st day, but it showed higher nuclear translocation on the 4th and 7th day. Scale bar = 20 µm.
Reference
- Tamiello, C., Bouten, C. V., & Baaijens, F. P. (2015). Competition between cap and basal actin fiber orientation in cells subjected to contact guidance and cyclic strain. Scientific reports, 5(1), 1-10.
Comment
Page 7, second line: "significantly high staining in overexpressed LMNA (OE) cells as compared" write "higher"
Author response:
Thank you very much for correcting us. We have already updated it and grammatically fixed it.
“LMNA cells which showed significantly higher staining in overexpressed LMNA (OE) cells as compared to knockdown LMNA (KD) and control (CTL)”
Comment
Some issues should be adressed during proof reading: "Lamin" was corrected in the abstract ("lamin"), but it is also still written in other sections of the manuscript starting with capital letter. Page 3 second line: there is a surplus blank within the brackets. Legend Figure 2: "The Protein" protein should not be written in capital letters. There are a couple of lacking blanks and minor grammar errors throughout the whole manuscript.
Author response:
We highly appreciate that you guided us throughout the process. We have addressed all those words you mentioned in your comment. We have also tried our best to fix other minors grammatically mistakes. As for as the surplus blanks are concerned, these are due to the journal template, and we will discuss them with the editor before final publication as we have tried it on our computer but could not fix it.